# A Concise Binomial Model for Nonlinear Creep-Fatigue Crack Growth Behavior at Elevated Temperatures

**DOI:** 10.3390/ma15020651

**Published:** 2022-01-15

**Authors:** Jianxing Mao, Zhixing Xiao, Dianyin Hu, Xiaojun Guo, Rongqiao Wang

**Affiliations:** 1Research Institute of Aero-Engine, Beihang University, Beijing 100191, China; maojx@buaa.edu.cn (J.M.); wangrq@buaa.edu.cn (R.W.); 2Beijing Key Laboratory of Aero-Engine Structure and Strength, Beihang University, Beijing 100191, China; 3School of Energy and Power Engineering, Beihang University, Beijing 100191, China; 18514568541@163.com; 4China Aviation Power Machinary Research Institute, Zhuzhou 412002, China; gxj608@163.com

**Keywords:** creep, fatigue, crack growth, fracture mechanism, nickel-based superalloys

## Abstract

The creep-fatigue crack growth problem remains challenging since materials exhibit different linear and nonlinear behaviors depending on the environmental and loading conditions. In this paper, we systematically carried out a series of creep-fatigue crack growth experiments to evaluate the influence from temperature, stress ratio, and dwell time for the nickel-based superalloy GH4720Li. A transition from coupled fatigue-dominated fracture to creep-dominated fracture was observed with the increase of dwell time at 600 °C, while only the creep-dominated fracture existed at 700 °C, regardless of the dwell time. A concise binomial crack growth model was constructed on the basis of existing phenomenal models, where the linear terms are included to express the behavior under pure creep loading, and the nonlinear terms were introduced to represent the behavior near the fracture toughness and during the creep-fatigue interaction. Through the model implementation and validation of the proposed model, the correlation coefficient is higher than 0.9 on ten out of twelve sets of experimental data, revealing the accuracy of the proposed model. This work contributes to an enrichment of creep-fatigue crack growth data in the typical nickel-based superalloy at elevated temperatures and could be referable in the modeling for damage tolerance assessment of turbine disks.

## 1. Introduction

Crack growth assessment on load-bearing components serving at elevated temperatures, e.g., the fracture-critical component turbine disk in an aeroengine, is one of the most crucial contents to ensure the structural integrity [1,2,3]. Temperature effect, especially aligned with the dwell of applied stress, contributes to alternating mechanisms of crack growth [4,5,6]. This thermal-mechanical coupling effect consequently increases the complexity in preventing critical fracture failures, remaining as one of the most attracting topics in high-temperature fracture mechanics [7,8,9].

Existing studies [4,10,11] revealed that the creep-fatigue crack growth (CFCG) behavior is significantly affected by temperature, applied stress, dwell time, and their conjoined effects. To date, a considerable number of investigations focusing on the mechanisms of crack growth have been carried out, supporting the construction of predicting models. Fatigue damage occurs as the material undergoes cyclic stress, while creep damage occurs when the material is subjected to sustained loading conditions. The total damage of creep-fatigue failure is determined by fatigue, creep, and their interaction. The proportion of each term depends on the environmental and loading condition, contributing to the main challenge of description on creep-fatigue crack growth behavior.

Basically, it is widely accepted that the time-dependent creep damage leads to intergranular crack growth, while the cycle-dependent fatigue damage contributes to transgranular crack growth. However, the truth turns out to be more complicated. Sadananda et al. revealed that the predominant fracture mode of Inconel-718 was intergranular at lower stress intensity levels, while transgranular at higher stress intensity levels (>60 MPa·m^0.5^) at 540 °C with 0 s and 60 s dwell time [12]. However, they found that transparent failure mode changed from transgranular to intergranular as the stress intensity level increases for Udmet-700 at 800 °C with 60 s dwell time [7]. Moreover, as for an identical DA Inconel-718 material, the fracture mode altered from transgranular to intergranular as temperature decreased from 700 °C to 550 °C when the local stress level exceeds the fracture toughness at the crack tip [8,13]. Therefore, it is of great importance to reveal the inherent mechanism for the alternate fracture mode depending on stress intensity levels at elevated temperatures.

The mechanism of crack growth affected by creep-fatigue interaction has been systematically investigated, where the form of creep damage varies significantly according to the level of stress intensity [5,14]. A general description can be established as (1) lower stress intensity levels: creep damage occurs at grain boundaries, where voids nucleation and growth dominate the crack growth behavior; (2) medium stress intensity levels: creep damage leads to crack blunting, while grain boundary separation motivates the crack propagation; and (3) higher stress intensity levels: creep damage is manifested by the nucleation of growth of voids in crystal, especially around inclusions or secondary phases, while the crack merging accelerates the crack growth rate.

Besides the creep damage mechanism, the crack-tip plastic zone size serves as another important factor that affects the creep-fatigue crack growth. As for a small cyclic plastic zone that is constrained to a limited number of grains, a shear-dominated crack growth occurs along the principal slip direction. When the cyclic plastic zone enlarges to the scale containing several grains, the crack growth is influenced by the double-slip mechanism. When the cyclic plastic zone derived from fatigue damage comes with the creep damage, the creep-fatigue interaction might accelerate or slow the crack growth, which depends on the dislocation slip at elevated temperatures [15,16], as well as the voids in grain boundaries [17] and secondary phases [18,19]. Therefore, the representation of time-dependent creep damage and cycle-dependent fatigue damage, and their interaction, becomes the main task in crack growth rate modeling for high-temperature structures.

The accurate description of crack growth behavior depends on two aspects: the first is the selection of dominant factor in fracture mechanics to correlate the stress-strain field around the crack tip, and the second is the establishment of proper format to relate the crack growth rate to the dominant factor. As for the small-scale yielding condition, the stress intensity factor K is always selected as the dominant factor. However, when the inelastic component of deformation is too large to be ignored, the usage of K is no more sufficient to describe the fracture process, leading to two kinds of approaches to ensure the accuracy of crack growth rate prediction. The first one is to develop a new factor to include the inelastic deformation, i.e., *C** for extensive creep conditions [20], and *C*(*t*) for transparent creep conditions [21]. To obtain an accurate estimation of *C** or *C*(*t*), visco-plastic constitutive models were developed to calculate the stress-stain field around the crack tip, as represented by the works of Landes and Begley [22], Goldman and Hutchinson [23]. Afterwards, damage parameters were introduced into the constitutive models to represent a more comprehensive creep state, such as the single damage variable models [24,25,26,27,28,29,30], double damage variable models [31,32], and triple damage variable models [33,34]. Although these *C**- or *C*(*t*)-based models were quite persuasive due to the explicit physical meaning of creep deformation, it has not been adopted in practical engineering applications because of the limitation of creep constitutive models.

Therefore, aiming at an accurate prediction in crack growth rate of materials at elevated temperatures, scholars have proposed several phenomenological models and different control parameters to elaborate the stress-strain field of crack tip. From a general standpoint, these models can be divided into three categories [35]. The first one was the mononomial model, which merely focused on a single failure mode, such as the fatigue crack growth (FCG). The second one was the binomial model based on linear superposition, which included both the effects from cycle- and creep-dependence. The third one was the trinomial model that added the interaction term to the binomial model, resulting in a more comprehensive and capable predictor for creep-fatigue crack growth. Due to the complex conjoined effects from high temperature, dwell time, and micro-mechanism of fracture, there are no universally accepted rules or analytical models for describing the creep-fatigue crack growth rates at high temperatures.

In this paper, the creep-fatigue crack growth behavior of the Ni-based superalloy GH4720Li is investigated at two temperatures and three dwell times. A series of fracture surface observations are conducted to identify the fracture mode. Typical mononomial, binomial and trinomial creep-fatigue crack growth models are then reviewed and applied, where sensitivity analysis is accordingly carried out. Aiming at ensuring a satisfied model applicability, we only consider the linear elastic fracture models that include modification terms to represent the load and environmental conditions. This paper presents a comprehensive summary on existing creep-fatigue crack growth models and could be referred for a proper choice of analysis models during damage tolerance assessment.

## 2. Material and Experiment

### 2.1. Material and Specimen

The material tested in this study is from a forged turbine disk of GH4720Li superalloy in the Chinese series (Central Iron & Steel Research Institute, Beijing, China), which is similar to the Udimet 720Li in the US series. The chemical content is listed in Table 1. During material preparation, solution treatment was firstly conducted at 1080–1110 °C for 2–4 h, followed by air cooling to room temperature; then a heat treatment was carried out at 760 °C for 8 h, with furnace cooling to 650 °C and preservation for 16 h. The microstructure is observed via a scanning electronic microscope (SEM) using Hitachi SU3500 (Hitachi Ltd, Tokyo, Japan), assisted by two etching methods following the surface polishing. A reagent of 10 g CuCl_2_, 100 mL HCl, and 100 mL C_2_H_5_OH is used for grain boundary observation (etching for 20–40 s), while a solution containing 150 mL H_3_PO_4_, 10 mL H_2_SO_4_, 15 g Cr_2_O_3_ is used for electro-polishing to expose the second phases (holding for 3–5 s at 3.5 V). The scanning electron (SE) images and backscattered electron (BSE) image in SEM observation at different scale reveal an average grain size of 10.565 μm (Weibull distribution is adopted in the statistical analysis, where λ = 11.928 and k = 2.327). The coarse blocky primary gamma precipitates (γ’) ranging from 1–4 µm mainly distribute on or near the grain boundaries, and the near spherical secondary γ’ precipitates with an average diameter of 0.5 µm randomly distribute within the grain. The fine spherical tertiary γ’ precipitates with a size smaller than 50 nm homogeneously disperse in the γ matrix. The detailed illustration was presented in our previous study [11].

The configuration of a typical contact tension (CT) specimen with W = 25 mm in width and B = 3.75 mm in thickness is adopted in the creep-fatigue crack growth experiments, as illustrated in Figure 1. Initial notches of 0.2 mm were created on all the specimens by wire-cut electro-discharge machining (EDM). All the specimens are cut from the rim region of the disk, with a diameter of 280 mm and a thickness of 64 mm. The disk is firstly treated by solid solution aging at 1080~1110℃ with a hold of 2~4 h, followed by oil quenching. Then, a stabilization heat treatment at 650℃ is conducted for 24 h, followed by air cool. At last, the disk is reheated to 760℃ for 16 h, followed by air cool.

### 2.2. Creep-Fatigue Crack Growth Experiment

The creep-fatigue crack growth tests are conducted on MTS Landmark^®^ Servohydraulic test systems (MTS, Eden Prairie, MN, USA), while the crack length is monitored by direct current potential drop (DCPD) method. The loading frequency for FCG with *t_h_* = 0 s is 10 Hz, while that for CFCG with *t_h_* = 10 s and 90 s is 0.33 Hz, i.e., a duration of 1.5 s for both the loading and unloading process. The investigated temperature (*T*), stress ratio (*R*), and dwell time (*t_h_*) are listed in Table 2, where two specimens are included for each tested condition. The crack growth data are presented in Figure 2.

Comparing the data of *da*/*dN*~Δ*K*, the effect of stress ratio, temperature, and dwell time can be accordingly analyzed. (1) By changing the stress ratio from 0.1 to 0.5, *da*/*dN* does not exhibit an obvious difference for conditions of *t_h_* = 0 s, i.e., the FCG cases. However, when it comes to the CFCG cases, *da*/*dN* generally increases with stress ratio at identical Δ*K.* (2) With an increase of temperature, the gap of *da*/*dN* between CFCG and FCG gets larger, which is more substantial with the enlongation of dwell time. (3) With the increase of dwell time, the gap between *da*/*dN* of CFCG and that of FCG gets larger, corresponding to the superposition of creep damage to fatigue damage.

### 2.3. Fractographic Analysis

A series of observation on fracture surface is carried out to investigate the transition of fracture mode along with the increase of dwell time, as illustrated in Figure 3, where the fracture surfaces of the specimen with *R* = 0.1 are shown. Those with *R* = 0.5 exhibit similar characteristics. Three locations are observed to investigate the fracture mechanism at different stages of CFCG, i.e., *a* = 6.75 mm for the initial stage, *a* = 9.25 mm for the midterm stage, and *a* = 11.75 mm for the later stage. At *T* = 600 °C with *t_h_* = 10 s, the fracture surface at the initial stage presents both creep and fatigue characteristics, where the dimples and striation platforms are marked in red and yellow, respectively. Then, it transit to the pure fatigue fracture at the midterm stage and the later stage. As *t_h_* increases to the 90 s, pure creep-dominated fracture exists at the initial stage and the midterm stage, while pure fatigue-dominated fracture is found at the later stage. Moreover, as the temperature increases to 700 °C, all the fracture surfaces exhibit typical creep-dominated dimples, regardless of the dwell time, which means the creep damage acts as the principal motivation for the crack growth.

## 3. Review of Existing Creep-Fatigue Crack Growth Models

### 3.1. Monomial Models

As for the monomial models for crack growth prediction, scholars generally focused on a single failure mode, i.e., fatigue crack growth. Additional terms and factors were included to involve the influence from temperature and loading frequency.

The most classical fatigue crack growth model is the Paris law [36], which describes the relation between crack growth rate *da*/*dN* and the stress intensity factor range ΔK by a simple power law formula. Due to the concise form of equation, it became the most widely used empirical prediction for fatigue crack growth rate in engineering applications. Further development of crack growth models were mostly based on the Paris law.
(1)$${\left(\frac{da}{dN}\right)}_{fatigue}=C{\left(\Delta K\right)}^{n}$$
where Δ*K* is the stress intensity factor range, *C* and *n* are the fitting parameters.

The Paris law presented a simple linear relation between *da*/*dN* and Δ*K* in a log-log diagram. The influence from loading condition and environmental temperature is attributed to the change of parameters *C* and *n*.

It was widely accepted that the stress ratio exhibited obvious influence on *da*/*dN*, where crack closure effect decreased effective Δ*K*, thereby lowered *da*/*dN*. To describe the influence from stress ratio, Walker equation [37] used a single linear segment to fit the data, which can be given by
(2)$${\left(\frac{da}{dN}\right)}_{fatigue}=C{\left[\frac{\Delta K}{{\left(1-R\right)}^{1-m’}}\right]}^{n}$$
where *R* is the stress ratio, *C*, *m*’, and *n* are the fitting parameters.

The Paris model and Walker model mainly concerned the second stage of crack growth and ignored the difference of the relation at the first stage (near threshold) and third stage (close to unstable fracture). Therefore, NASA Johnson Space Center, Air Force Research Laboratory, and Southwest Research Institute developed a more comprehensive model to consider the influence from stress ratio and nonlinear stages [38], as
(3)$${\left(\frac{da}{dN}\right)}_{fatigue}=C{\left(\frac{1-f_{N}}{1-R}\Delta K\right)}^{n}\frac{{\left(1-\frac{\Delta K_{th}}{\Delta K}\right)}^{p}}{{\left(1-\frac{K_{max}}{K_{c}}\right)}^{q}}$$
where *f_N_* is the Newman crack closure function, *R* is the stress ratio, *K_th_* and *K_c_* are the crack growth threshold and fracture toughness respectively, *C*, *n*, *p*, *q* are the fitting parameters. The function *f_N_* can be further represented by [39]
(4)$$\begin{array}{l} f_{N}=A_{0}+A_{1}R+A_{2}R^{2}+A_{3}R^{3}\\ A_{0}=\left(0.825-0.34\alpha +0.05\alpha^{2}\right){\left[\cos\left(\frac{\pi S_{max}}{2\sigma_{0}}\right)\right]}^{1/\alpha }\\ A_{1}=\left(0.415-0.071\alpha \right)\frac{S_{max}}{\sigma_{0}}\\ A_{2}=1-A_{0}-A_{1}-A_{3}\\ A_{3}=2A_{0}+A_{1}-1 \end{array}$$
where *S_max_* is the maximum nominal stress, *σ*_0_ is the flow stress, which is the average of the yield stress *σ_s_* and tensile strength *σ_b_*, *α* is the factor to represent plane stress (*α* = 1) or plane strain state (*α* = 3).

### 3.2. Binomial Models

In order to describe the creep-fatigue crack growth, scholars firstly assumed that the creep and fatigue components stayed independent to the other. Therefore, the binomial models based on a linear superposition relation was proposed. A general expression of these models can be written as
(5)$${\left(\frac{da}{dN}\right)}_{creep-fatigue}={\left(\frac{da}{dN}\right)}_{fatigue}+{\left(\frac{da}{dN}\right)}_{creep}$$
where the term (*da*/*dN*) fatigue can still be represented as previously introduced monomial models, while the term (*da*/*dN*) creep is related to parameters as stress intensity range Δ*K*, temperature *T*, stress ratio *R*, dwell time *t_h_*, and frequency *f* [40,41].

Magnus et al. [42] proposed that the creep term for crack growth at elevated temperatures can be described by *K_max_* and *t_h_* in a format similar to the Paris law, and indicated that the crack growth rate fitted the linear superposition of fatigue and creep terms in different environments and temperatures. As enlightened by the Paris law, *da*/*dN* for creep-fatigue crack growth can be given by
(6)$${\left(\frac{da}{dN}\right)}_{creep-fatigue}=C{\left(\Delta K\right)}^{n}+A{\left(K_{max}\right)}^{m}\cdot t_{h}$$
where *C*, *n*, *A*, *m* are the fitting parameters.

### 3.3. Trinomial Models

The linear superposition binomial models assumed an independent creep and fatigue damage; however, subsequent experimental studies further identified an interacting type of damage as the dwell time increased. In this regard, trinomial models were developed with an additional term to represent the creep-fatigue interaction. A general expression of the trinomial models can be written as
(7)$${\left(\frac{da}{dN}\right)}_{creep-fatigue}={\left(\frac{da}{dN}\right)}_{fatigue}+{\left(\frac{da}{dN}\right)}_{creep}+{\left(\frac{da}{dN}\right)}_{interaction}$$
where (*da*/*dN*)*_interaction_* is the interaction term to include the combined effect of environment and load condition.

Byrne et al. [43] conducted creep-fatigue crack growth experiments on Waspaloy alloy. By using the Paris law to describe the fatigue crack growth rate, and a creep term consisted of Δ*K*, *R*, and *t_h_*, *da*/*dN* was expressed as
(8)$${\left(\frac{da}{dN}\right)}_{creep-fatigue}=C{\left(\Delta K\right)}^{n}+A{\left(K_{max}\right)}^{m}t_{h}+A{\left(K_{max}\right)}^{m}t_{h}\left[\frac{1-R^{m+1}}{1-R}\frac{1}{f_{0}}\left(m+1\right)\right]$$
where *f*_0_ is a reference frequency, *C*, *n*, *A*, *m* are the fitting parameters.

As enlightened by Saxena’s works [44], Liu et al. [35] presented a new format of expression for the time-dependent damage component, and realized the model validation on a set of materials, such as Ni-based superalloy FGH97, Aluminum alloy 2650, P91 steel, and Ni-based superalloy Astroloy.
(9)$${\left(\frac{da}{dN}\right)}_{creep-fatigue}=C{\left(\Delta K\right)}^{n}+A{\left(K_{max}\right)}^{m}t_{h}+A{\left(K_{max}\right)}^{m}t_{h}\beta \exp\left[-\frac{1}{2}{\left(\ln\frac{t_{h}}{t_{inc}}\right)}^{2}\right]$$
where *C*, *n*, *A*, *m*, *β* are fitting parameters, *t_inc_* is the incubation time to represent the characteristic value of the most significant level of creep-fatigue interaction effect.

## 4. Model Implementation and Discussions

As lightened by above works, a unified trinomial model can be established as
(10)$${\left(\frac{da}{dN}\right)}_{creep-fatigue}=C{\left(f_{1}\left(R\right)\cdot \Delta K\right)}^{n}\cdot \frac{f_{2}\left(\Delta K,\Delta K_{th}\right)}{f_{3}\left(\Delta K,K_{c}\right)}+A{\left(K_{max}\right)}^{m}t_{h}\left[1+f_{4}\left(R,t_{h}\right)\right]$$
where function *f*_1_, *f*_2_, *f*_3_, and *f*_4_ can be expressed according to Section 3, which are summarized in Table 3.

A stepwise validation on each function will be evaluated individually with the data achieved from experiments, as shown in following sections.

### 4.1. Linear Part for Pure Fatigue

Function *f*_1_ describes the dependence of *da*/*dN* on stress ratio *R*, at the stage II of fatigue crack growth. By transforming to a log-log format, the Walker model and NASGRO model can be expressed as
(11)$$\mathrm{Walker}\;\mathrm{model}:\;\mathrm{lg}\left[{\left(\frac{da}{dN}\right)}_{fatigue,\;stage\;II}\right]=\mathrm{lg}C+n\mathrm{lg}\Delta K+n\mathrm{lg}\left[{\left(1-R\right)}^{m’-1}\right]$$
(12)$$\mathrm{NASGRO}\;\mathrm{model}:\;\mathrm{lg}\left[{\left(\frac{da}{dN}\right)}_{fatigue,\;stage\;II}\right]=\mathrm{lg}C+n\mathrm{lg}\Delta K+n\mathrm{lg}\left(\frac{1-f_{N}}{1-R}\right)\;$$

Accordingly, the difference between *da*/*dN* at *R* = 0.1 and *R* = 0.5 can be expressed as
(13)$$\mathrm{Walker}\;\mathrm{model}\;:\mathrm{lg}\left[\frac{{\frac{da}{dN}|}_{R=0.1}}{{\frac{da}{dN}|}_{R=0.5}}\right]=n\mathrm{lg}\left[\frac{{{\left(1-R\right)}^{m’-1}|}_{R=0.1}}{{{\left(1-R\right)}^{m’-1}|}_{R=0.5}}\right]=n\left(m’-1\right)\mathrm{lg}\left(1.8\right)$$
(14)$$\mathrm{NASGRO}\;\mathrm{model}\;:\mathrm{lg}\left[\frac{{\frac{da}{dN}|}_{R=0.1}}{{\frac{da}{dN}|}_{R=0.5}}\right]=n\mathrm{lg}\left[\frac{{\frac{1-f_{N}}{1-R}|}_{R=0.1}}{{\frac{1-f_{N}}{1-R}|}_{R=0.5}}\right]=n\mathrm{lg}\left(\frac{1}{1.8}\frac{{1-f_{N}|}_{R=0.1}}{{1-f_{N}|}_{R=0.5}}\right)\;$$

By selecting the crack growth data at stage II under pure fatigue load, the differences of *da*/*dN* at *R* = 0.1 and *R* = 0.5 at elevated temperature are illustrated in Figure 4. Figure 4a illustrates an intersecting type of *da*/*dN* at *R* = 0.1 and *R* = 0.5 at *T* = 600℃. Before the intersecting point, *da*/*dN* at *R* = 0.1 is lower than that at *R* = 0.5. On the other hand, Figure 4b exhibits a parallel linear fit for different stress ratios, while the ratio of (*da*/*dN*)*_R_*_=0.5_/(*da*/*dN*)*_R_*_=0.1_ is approximate to 1 with variant Δ*K*.

As revealed by the Walker model and NASGRO model, cf. Equations (13) and (14) respectively, the ratio of (*da*/*dN*)*_R_*_=0.5_/(*da*/*dN*)*_R_*_=0.1_ stays constant, which means the stress ratio only influence the intercept of *da*/*dN* with Δ*K* in log-log diagram, which conflicts with the achieved data in Figure 4. Therefore, a new format of fatigue crack growth model involving the effect of stress ratio should be established. By adopting the Paris model represented by Equation (1), the fitting parameters can be determined, as illustrated in Table 4.

### 4.2. Nonlinear Part for Pure Fatigue

The functions *f*_2_ and *f*_3_ in Table 3 represent the nonlinear parts (stage I and III) of fatigue crack growth, which is dominated by Δ*K_th_* and *K_c_*. The process to deal with the nonlinear parts is identical. Practically, since the obtained data does not include the nonlinear part at stage I, we only consider that at stage III in this section while setting function *f*_2_ = 1.

By dividing the fatigue crack growth data with the established model for stage II, i.e., the Paris model with parameters in Table 4, the value of function *f*_3_ on each data point can be accordingly determined, as illustrated in Figure 5. Through a fitting process, parameter *K_c_* and *q* can be determined, by which the fitting curves are marked blue. However, the original term in the NASGRO model underestimates the effect of *f*_3_ as *K_max_* approaches *K_c_*. Therefore, a new format of *f*_3_ is constructed, as enlightened by the NASGRO model, which can be expressed as
(15)$$f_{3}={\left[1-{\left(\frac{K_{max}}{K_{c}}\right)}^{q_{1}}\right]}^{q_{2}}$$

The fitting results for original term and proposed term are listed in Table 5.

### 4.3. Parts for Creep and Creep-Fatigue Interaction

After the determination of *da*/*dN* for pure fatigue, we can realize the evaluation of *da*/*dN* for creep and interaction through a transformation of Equation (10), as expressed by Equation (16), while the corresponding data are shown in Figure 6.
(16)$$\begin{array}{l} {\left(\frac{da}{dN}\right)}_{creep}+{\left(\frac{da}{dN}\right)}_{interaction}\\ ={\left(\frac{da}{dN}\right)}_{creep-fatigue}-C_{R}{\left(\Delta K\right)}^{n_{R}}\cdot {\left\{1-{\left[\frac{\Delta K}{\left(1-R\right)K_{c}}\right]}^{q_{1}}\right\}}^{-q_{2}}\\ =A{\left(K_{max}\right)}^{m}t_{h}\left[1+f_{4}\left(R,t_{h}\right)\right] \end{array}$$
where *C_R_* and *n_R_* are the parameters *C* and *n* in the Paris model at an arbitrary stress ratio *R*, which can be obtained by a linear interpolation as
(17)$$\mathrm{lg}\left(C_{R}\right)=\frac{R-0.1}{0.5-0.1}\mathrm{lg}\left(\frac{C_{0.5}}{C_{0.1}}\right)+\mathrm{lg}\left(C_{0.1}\right)$$
(18)$$n_{R}=\frac{R-0.1}{0.5-0.1}\left(n_{0.5}-n_{0.1}\right)+n_{0.1}$$

From the data illustrated in Figure 6, facts can be found as (1) the value of *da*/*dN* for creep and interaction increases with *t_h_* at identical *T* and *R*, and (2) the slop of *da*/*dN* to Δ*K* varies with *R*, which is more prominent at *T* = 700 °C While retrospecting to the candidate formats of *f*_4_, the Byrne model integrates *R* on the interaction part, without changing the slop of *da*/*dN* to Δ*K*. Meanwhile, this model cannot reflect the dependence of *da*/*dN* on *t_h_*. On the other hand, the Liu model perfectly fits the requirement of the description on *da*/*dN* for creep and interaction, while assuming that *A* and *m* are two constants depending on *R*. It can be expressed as
(19)$${\left(\frac{da}{dN}\right)}_{creep}+{\left(\frac{da}{dN}\right)}_{interaction}=A{\left(K_{max}\right)}^{m}\cdot t_{h}\left\{1+\beta \exp\left[-\frac{1}{2}{\left(\ln\frac{t_{h}}{t_{inc}}\right)}^{2}\right]\right\}$$

The next step is to determine the incubation time *t_inc_*. In our previous study [11], a series of experiments were conducted to investigate the interaction of creep and fatigue in the GH4720Li superalloy at *T* = 650 °C, *σ* = 1000 MPa, and *R* = 0.1. Life results in cycle and second are listed in Table 6, where *N* is the life in cycle, *t* is the life in second, *D_cr_* is the fraction of creep damage by dividing *t* of creep-fatigue by that of pure creep, *D_in_* = 1 − *D_cr_* is the fraction of interaction damage where the pure fatigue damage is ignored due to the difference of magnitude of level. By using the data of *D_in_*/*D_cr_* to *t_h_*, the material parameters in term *f*_4_ in the Liu model can be determined, as *β* = 2.2685 and *t_inc_* = 180 s in Equation (19). Then, material parameters *A* and *m* can be subsequently fitted, on the basis of the data in Figure 6, which are listed in Table 7.

### 4.4. Model Implementation and Validation

Based on Section 4.1, Section 4.2 and Section 4.3, the reconstructed model can be expressed as
(20)$$\begin{array}{ll} {\left(\frac{da}{dN}\right)}_{creep-fatigue} & =C{\left(\Delta K\right)}^{n}\cdot {\left[1-{\left(\frac{K_{max}}{K_{c}}\right)}^{q_{1}}\right]}^{-q_{2}}\\ & +A{\left(K_{max}\right)}^{m}\cdot t_{h}\left\{1+\beta \exp\left[-\frac{1}{2}{\left(\ln\frac{t_{h}}{t_{inc}}\right)}^{2}\right]\right\} \end{array}$$
where parameters *C*, *n*, *K_c_*, *q*_1_, *q*_2_, *A*, *m*, *β*, and *t_inc_* are given in Table 4, Table 5, and Table 7. Using the established model, the *da*/*dN* at investigated temperatures and stress ratios can be accordingly calculated, which is illustrated in Figure 7. In order to evaluate the accuracy of the proposed model, the correlation coefficient is adopted, of which the results are shown in Table 8.

Ten out of twelve in the calculated correlation coefficients are higher than 0.9, revealing the accuracy of the established crack growth model. By focusing on the predictions with correlation coefficients lower than 0.9, we can find that the mismatches of slop are derived from the variance in slop of the (*da*/*dN*)_creep_ + (*da*/*dN*)*_interaction_* data illustrated in Figure 6. Since we employ Equation (19) to describe the *da*/*dN* for creep and interaction, the term *A*(*K_max_*)*^m^* defines an identical slop for approximation. Efforts are still needed in future works to analyze the inconsistency of slop.

## 5. Conclusions

In summary, a concise binomial model for creep-fatigue crack growth is constructed on the basis on existing models, where the linear and nonlinear behaviors at different environmental and loading conditions are elaborated. The main conclusions can be drawn as follows.

(1)A series of creep-fatigue crack growth tests are carried out under different temperatures, stress ratios, and dwell times, where both linear and nonlinear relations are observed on the *da*/*dN*~Δ*K* data.(2)Fractographic analysis is carried out on the fractured specimens of each tested condition, where the fatigue-dominated fracture transits to the creep-dominated fracture as the dwell time increases from 10 s to 90 s at 600 °C, while the pure creep-dominated fracture is found at 700 °C.(3)A concise binomial model is constructed on the basis of existing models, where the linear terms are included to express the behavior under pure creep loading, and the nonlinear terms are introduced to represent the behavior near the fracture toughness and during the creep-fatigue interaction.(4)Through the model implementation and validation of the proposed model, the correlation coefficient is higher than 0.9 on ten out of twelve sets of experimental data, revealing the accuracy of the proposed model.

This work contributes to an enrichment of creep-fatigue crack growth data in the typical nickel-based superalloy at elevated temperatures and could be referable in the modeling for damage tolerance assessment of turbine disks.

## Figures and Tables

**Figure 1 materials-15-00651-f001:**
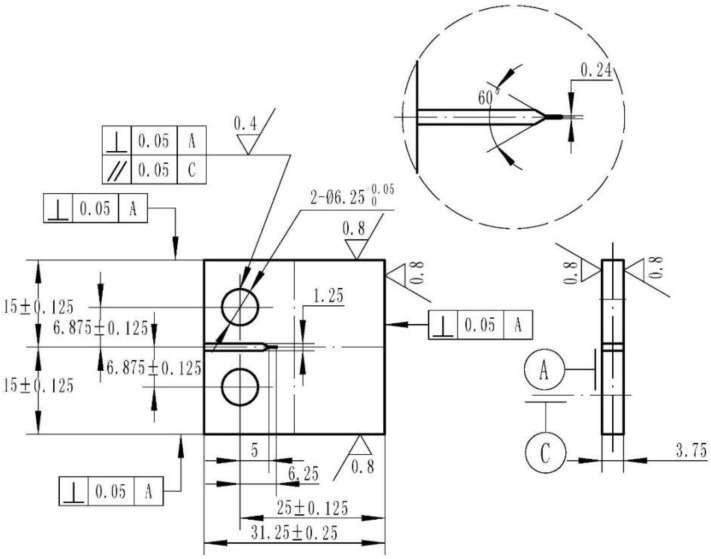
Geometry of specimen adopted in creep-fatigue crack growth test. (unit in mm).

**Figure 2 materials-15-00651-f002:**
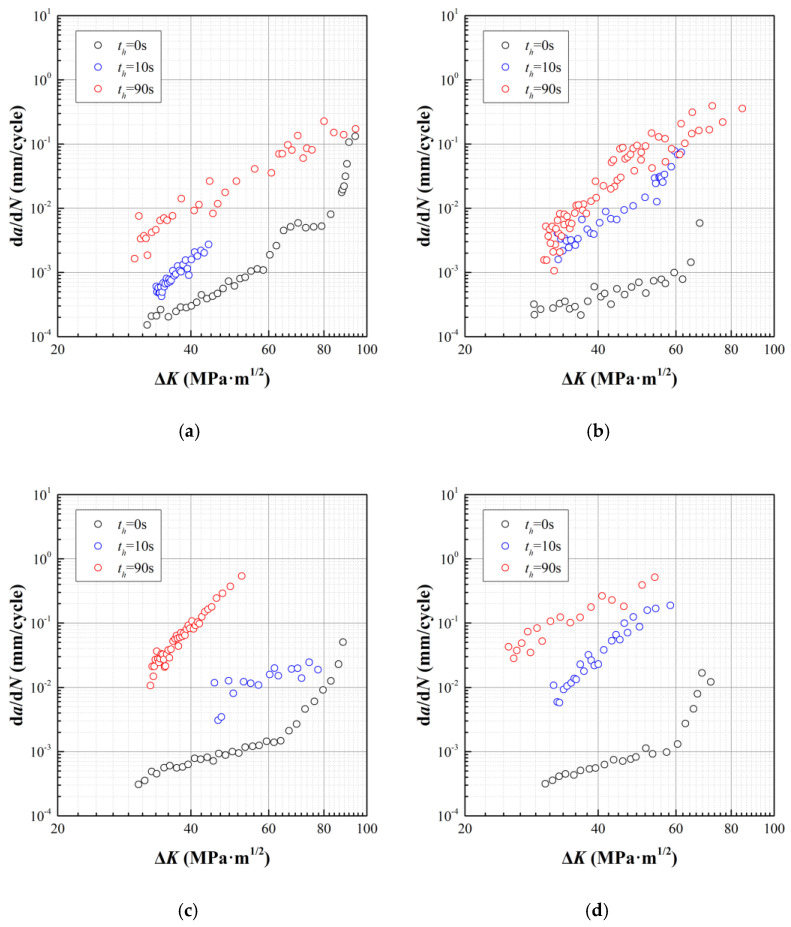
Experimental results of creep-fatigue crack growth at elevated temperatures. (**a**) *T* = 600 °C, *R* = 0.1; (**b**) *T* = 600 °C, *R* = 0.5; (**c**) *T* = 700 °C, *R* = 0.1; (**d**) *T* = 700 °C, *R* = 0.5.

**Figure 3 materials-15-00651-f003:**
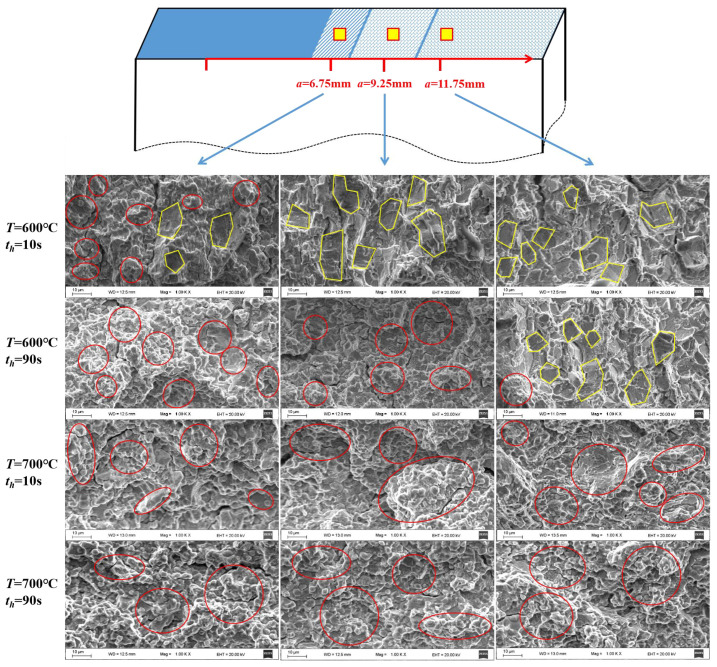
Fractographic observation at different locations with different dwell time and temperature.

**Figure 4 materials-15-00651-f004:**
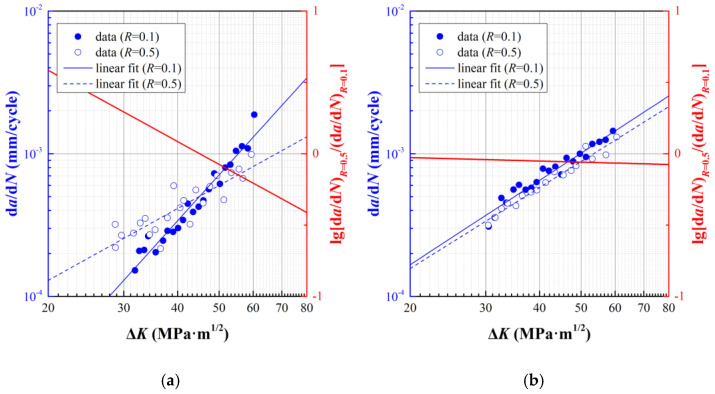
Experimental data and fitting results for fatigue crack growth at stage II. (**a**) *T* = 600 °C; (**b**) *T* = 700 °C.

**Figure 5 materials-15-00651-f005:**
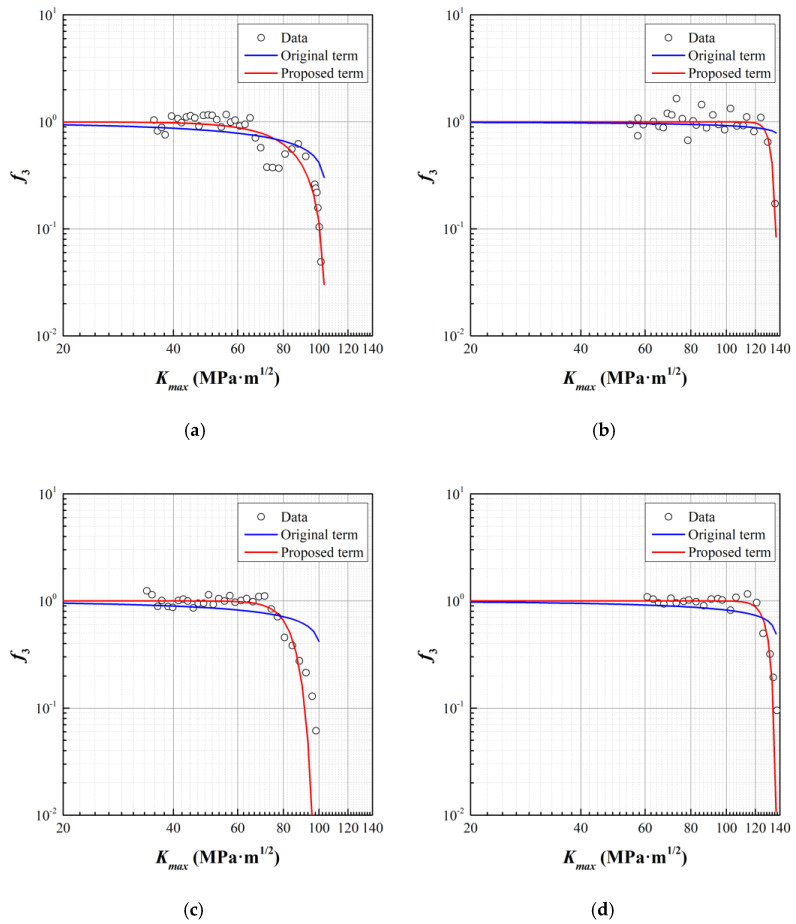
Experimental data and fitting results for the original and proposed *f*_3_. (**a**) *T* = 600 °C, *t_h_* = 10 s; (**b**) *T* = 700°C, *t_h_* = 10 s; (**c**) *T* = 700°C, *R* = 0.1; (**d**) *T* = 700°C, *R* = 0.5.

**Figure 6 materials-15-00651-f006:**
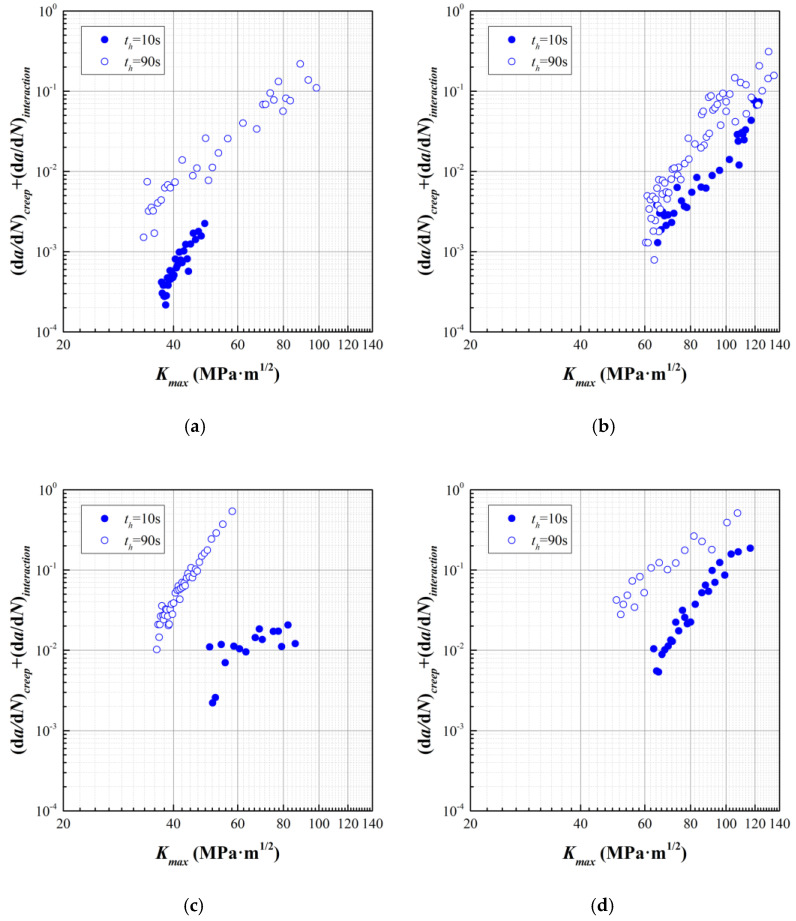
*da*/*dN* for creep and interaction calculated by Equation (16). (**a**) *T* = 600 °C, *R* = 0.1; (**b**) *T* = 700 °C, *R* = 0.5; (**c**) *T* = 700 °C, *R* = 0.1; (**d**) *T* = 700 °C, *R* = 0.5.

**Figure 7 materials-15-00651-f007:**
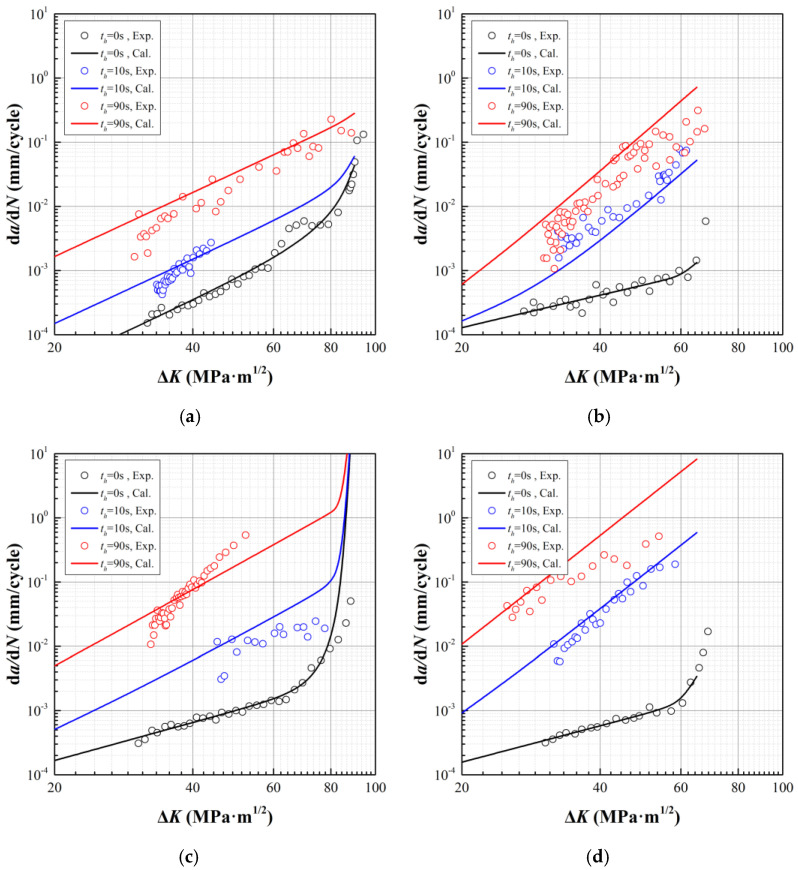
Experimental and predicted *da*/*dN* under pure fatigue and creep-fatigue. (**a**) *T* = 600 °C, *R* = 0.1; (**b**) *T* = 700 °C, *R* = 0.5; (**c**) *T* = 700 °C, *R* = 0.1; (**d**) *T* = 700 °C, *R* = 0.5.

**Table 1 materials-15-00651-t001:** Chemical content of investigated GH4720Li (in wt%).

Ni	C	Si	Mn	Cr	Co	Mo	Al
Balance	0.01–0.02	≤0.2	≤0.15	15.5–16.5	14–15.5	2.75–3.25	2.25–2.75
Fe	B	Zr	P	Cu	Ti	W	
≤0.05	0.01–0.02	0.025–0.05	≤0.015	≤0.01	4.75–5.25	1–1.5	

**Table 2 materials-15-00651-t002:** Investigated conditions in creep-fatigue crack growth experiments.

*T* (°C)	*R*	*P_max_* (kN)	*t_h_* (s)
600	0.1	4	0, 10, 90
	0.5	6	0, 10, 90
700	0.1	4	0, 10, 90
	0.5	6	0, 10, 90

**Table 3 materials-15-00651-t003:** Crack growth models summarized in a unified format of Equation (10).

Model	*f* _1_	*f* _2_	*f* _3_	*f* _4_
Paris	1	1	1	0
Walker	${\left(1-R\right)}^{m’-1}$	1	1	0
NASGRO	$\frac{1-f_{N}}{1-R}$	${\left(1-\frac{\Delta K_{th}}{\Delta K}\right)}^{p}$	${\left(1-\frac{K_{max}}{K_{c}}\right)}^{q}$	0
Magnus	1	1	1	0
Byrne	1	1	1	$\frac{1-R^{m+1}}{1-R}\frac{1}{f_{0}}\left(m+1\right)$
Liu	1	1	1	$\beta \exp\left[-\frac{1}{2}{\left(\ln\frac{t_{h}}{t_{inc}}\right)}^{2}\right]$

**Table 4 materials-15-00651-t004:** Fitting parameters for the Paris model.

*T* (°C)	*R*	lg(*C*)	*C*	*n*
600	0.1	−8.8054	1.5653 × 10^−9^	3.3299
	0.5	−6.0641	8.6278 × 10^−7^	1.6726
700	0.1	−6.3361	4.6121 × 10^−7^	1.9659
	0.5	−6.2592	5.5055 × 10^−7^	1.8862

**Table 5 materials-15-00651-t005:** Fitting parameters for the original and proposed *f*_3_.

*T* (°C)	*R*	Original *f*_3_			Proposed *f*_3_			
		*K_c_* (MPa · m^1/2^)	*q*	*RMSE*	*K_c_* (MPa · m^1/2^)	*q* _1_	*q* _2_	*RMSE*
600	0.1	102	0.28	0.23	105	4.27	1.28	0.16
	0.5	136	0.06	0.26	140	34.0	4.27	0.21
700	0.1	99	0.22	0.23	102	11.6	6.86	0.10
	0.5	138	0.15	0.21	138	22.1	2.75	0.09

RMSE refers to the root mean square error.

**Table 6 materials-15-00651-t006:** Life results for pure fatigue, creep-fatigue, and pure creep.

Type	*t_h_* (s)	*N* (Cycle)	*t* (s)	*D_cr_*	*D_in_*	*D_in_*/*D_cr_*
Fatigue	0	7 × 10^6^	-	0	0	-
Creep-fatigue	180	397	71,460	0.3054	0.6946	2.2744
	720	177	127,440	0.5446	0.4554	0.8362
	1800	102	183,600	0.7846	0.2154	0.2745
	2700	80	216,000	0.9231	0.0769	0.0833
Creep	∞	-	234,000	1	0	0

**Table 7 materials-15-00651-t007:** Fitting parameters for the terms of creep and interaction.

*T* (°C)	*R*	*A*	*m*	*β*	*t_inc_* (s)
600	0.1	1.8863 × 10^−10^	3.3241	2.2685	180
	0.5	1.9603 × 10^−16^	6.2007		
700	0.1	7.2236 × 10^−11^	3.9842		
	0.5	3.4548 × 10^−14^	5.6390		

**Table 8 materials-15-00651-t008:** Correlation coefficient between calculated and experimental *da*/*dN*.

*T* (°C)	*R*	*t_h_* (s)		
		0	10	90
600	0.1	0.977	0.957	0.902
	0.5	0.995	0.951	0.826
700	0.1	0.911	0.766	0.978
	0.5	0.983	0.946	0.955

## Data Availability

The data presented in this study are available on request from the corresponding author.

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
