# Peer review of "A Concise Binomial Model for Nonlinear Creep-Fatigue Crack Growth Behavior at Elevated Temperatures"

_materials, 2022, doi:10.3390/ma15020651_

Round 1
Reviewer 1 Report
Dear Authors,
Please see the attached review comments.

Reviewer 2 Report
A few minor things need to correct
Line 133-4: The authors had mentioned the different name of the investigated superalloy. In this paper, the authors named the alloy called GH4720Li superalloy, but the previous paper had given GH720Li superalloy. In spite of both alloys have the same chemical composition, could the authors clarify this confused issue?
Line151: “The crack 151 growth data are presented in Figure 2.” This is referred to results section. Results section is also missing?
Line 158: “2.3 Fractographic analysis” this is relating to result section.
Line 169: the scale bar length in Figure 3a is different from others, namely Figure 3b,c, and d. WHY?
Lines 161, 165: Figure e missing?
Line 220: [refs] missing?
Line 247, 249 and 303: Figure x?
Line 257: the fitting parameters determined and illustrated in Table 3. Could the authors compare it to other superalloys?
Line 267-273: The paragraph needs to clarify, in particular, how q1, q2 and Kc for equation 15 or proposed f3 were been estimating.
Line 286: Figure 6. …calculated by equation 21. Equation 16, isn’t it?
Line 303: Equation 19 does not exist?
Line 305 and line 307: Table 5 and Table 6 should give which equation the fitting parameters were been estimated.
Author Response
A few minor things need to correct.
- Line 133-4: The authors had mentioned the different name of the investigated superalloy. In this paper, the authors named the alloy called GH4720Li superalloy, but the previous paper had given GH720Li superalloy. In spite of both alloys have the same chemical composition, could the authors clarify this confused issue?
Reply: The name GH720Li is an old version of this material. In the newest material handbook, it has been changed to GH4720Li in the Chinese superalloys system.
- Line151: “The crack growth data are presented in Figure 2.” This is referred to results section. Results section is also missing?
Reply: This paper is constructed by experimental studies and numerical modeling, including crack growth test, fractographic analysis, as well as a stepwise process of application on existing models and corresponding modification. If we choose to have a individual section for results, the readers might have some difficulties to align the methods to the corresponding results. Thereby, the authors suggest to present the result following the description of each method.
- Line 158: “3 Fractographic analysis” this is relating to result section.
Reply: Similar to the previous comment, the authors suggest to present the result following the description of each method to make it easier to be comprehended by the reader.
- Line 169: the scale bar length in Figure 3a is different from others, namely Figure 3b,c, and d. WHY?
Reply: The aim to present Figure 3a is to exhibit the striations for fatigue damage, while that for Figure 3b and c is to show the typical dimples for creep damage. Due to the difference in length scale, the authors use different scale bar for a proper illustration.
- Lines 161, 165: Figure e missing?
Reply: Thanks for pointing out this mistake. Initially the authors use three figures to exhibit the different stage of fracture surface of each experimental condition. However, since the achieved data mostly belongs to the second stage of crack growth, the authors choose to use the corresponding part of fracture surface to illustrate the dominate mechanism, which changes from fatigue-dominated to creep-dominated with the increase of temperature and dwell time. The context has been accordingly revised.
- Line 220: [refs] missing?
Reply: Thanks for pointing out this mistake. It has been removed since all the required references have been mentioned above.
- Line 247, 249 and 303: Figure x?
Reply: Thanks for pointing out this mistake. They have been corrected in the revised version.
- Line 257: the fitting parameters determined and illustrated in Table 3. Could the authors compare it to other superalloys?
Reply: This work is undergoing on the superalloy FGH96 at this stage. However, since we focus on the superalloy GH4720Li in this paper, the author prefer to present these data elsewhere in the future.
- Line 267-273: The paragraph needs to clarify, in particular, how q1, q2 and Kc for equation 15 or proposed f3 were been estimating.
Reply: The achievement of q1, q2 and Kc is realized by a nonlinear fitting using matlab code. Since this is a common approach to determine the parameters in functions, the authors do not present it in details.
- Line 286: Figure 6. …calculated by equation 21. Equation 16, isn’t it?
Reply: Thanks for pointing out this mistake. They have been corrected in the revised version.
- Line 303: Equation 19 does not exist?
Reply: Thanks for pointing out this mistake. The number of equation has been corrected in the revised manuscript.
12) Line 305 and line 307: Table 5 and Table 6 should give which equation the fitting parameters were been estimated.
Reply: The data in Table 5 and Table 6 is used to determine the parameters in Eq.(19). This is mentioned in the last sentence in the paragraph before Table 5.
Reviewer 3 Report
This paper investigated the creep-fatigue crack growth behavior of a nickel based superalloy at elevated temperatures. A concise binomial crack growth model was established on the basis of existing phenomenal for prediction of the crack growth rate under creep-fatigue condition. The proposed model is significant for the damage tolerance assessment of structures that are subjected to creep and fatigue under elevated temperatures. The paper is well structured and nicely presented. However, before the publication of this paper, there are several shortages must to be overcomed.
- Some grammatical and formatting errors.
Line 88 “To obtained an accurate estimation”.
Line 91 “Hutchinson[23], Afterwards,”.
Line 174 “generally stayed focused on a”.
Line 220 “to represent the creep-fatigue interaction[refs].”.
Line 247 “are illustrated in Figure x. Figure x(a)”
Line 303 ” m in Eq.(19) can be accordingly fitted, on the basis of the data in Figure x,”
- Missing Figure 3(e) in line 165? Missing Eq.(19) in line 303?
- Please provide specific fatigue load and frequency under different stress ratios in section 2.2.
- More analyses should be provided for interpretation of the crack growth data in Figure 2. Specifically, the effects of stress ratio, temperature and dwell time on da/dN need to be clarified.
- Please point out the dimples and fatigue striations in Figure 3. A higher magnification of SEM image is helpful for demonstrating the existence of fatigue striations. Please also show the crack growth direction.
- Are these SEM images (Figure 3) of fracture surfaces observed at the same crack length or stress intensity factor range under different test conditions?
- I recommend for providing more SEM photos of fracture surfaces under different temperatures, stress ratios and dwell times for supporting your second conclusion, i.e. the transition of the fracture mode, since you have done a lot of fracture surface observations.
- Please give a thorough explanation on the errors between the experimental value and predicted value by the proposed model, since the correlation coefficients under some specific conditions in Table 7 are lower than 0.9.
Round 2
Reviewer 1 Report
Dear Authors,
Thank you for responding to all review comments well.
Author Response
Thanks for the reviewer for providing such valuable comments.
Reviewer 3 Report
In Section 2.3, you mentioned there were both intergranular and transgranular characteristics including a large number of intergranular dimples on the fracture surface. Intergranular fracture occurs along the grain boundary. However, from your SEM images, it is hard to find obvious evidences for Intergranular fracture. The results show a transition from fatigue-dominated to creep-dominated fracture. However, the fractures of your samples are still transgranular in nature for all conditions.
Author Response
Thanks for the comments. The authors agree with the reviewer, and have changed the statement of transgranular and intergranular to fatigue-dominated fracture and creep-dominated fracture. Details can be found in Section 2.3 in the uploaded revised version. Corresponding modifications have been made to the Abstract and Conclusion.
